# Perspectives on New Vaccines against Arboviruses Using Insect-Specific Viruses as Platforms

**DOI:** 10.3390/vaccines9030263

**Published:** 2021-03-16

**Authors:** Valéria L. Carvalho, Maureen T. Long

**Affiliations:** 1Department of Comparative, Diagnostic, and Population Medicine, College of Veterinary Medicine, University of Florida, 1945 SW 16th Ave, Gainesville, FL 32608, USA; 2Department of Arbovirology and Hemorrhagic Fevers, Evandro Chagas Institute, Ministry of Health, Rodovia BR-316, Km 7, s/n, Ananindeua, 67030-000 Para, Brazil

**Keywords:** vaccines, insect-specific virus, arbovirus

## Abstract

Arthropod-borne viruses (arboviruses) are global pathogens circulating endemically with local explosive outbreaks and constant encroachment into new locations. Few vaccines against arboviruses exist; most for humans are in development or clinical trials. Insect-specific viruses (ISVs) offer a unique platform for expression of arbovirus proteins, through the creation of ISV/arbovirus chimeras. Studies have shown promising results of these vaccines with several advantages over their wild-type counterparts. In this review, we discuss the current status of these potential vaccines using ISVs.

## 1. Introduction

According to the World Health Organization (WHO), vaccination provides the most effective prophylaxis against infectious disease. Notwithstanding anti-vaccination movements, vaccines are recognized as important and necessary tools to fight the advancement of diseases. Vaccination is a life-saving tool is now recognized by the non-medical public due the rise of COVID-19, and the global community is more accepting of vaccination as a way forward to resumption of healthy lives and economic wellbeing. Even without the contribution of SARS-CoV-2 disease vaccination, the WHO estimates that vaccinations save three million lives per year [1].

There are different types of vaccine technologies, beginning with whole pathogen inactivated and attenuated vaccines to novel designs composed of nucleic acids, virus-like particles, and conjugate vaccines. Each strategy has its own strengths and weaknesses [2,3]. For example, inactivated and attenuated whole pathogen vaccines tend to be highly immunogenic with safety issues, while engineered vaccines are considered safer but often induce lower immune responses [2,3].

Even though the world has advanced in vaccine technologies in recent decades, new vaccines are still yet unavailable to prevent many diseases, especially vector-borne viruses which are responsible for causing almost 20% of the global disease burden. While Dengue virus (DENV) alone was estimated to cause 96 million symptomatic cases/yr, infections such as Zika virus (ZIKV), West Nile virus (WNV) and Chikungunya virus (CHIKV) have long term consequences that contribute to this burden. There is no specific treatment to these diseases; and, few vaccines are available to prevent the establishment of the disease [4,5].

A recently discovered group of viruses also carried by mosquitoes called insect-specific viruses (ISVs), which are unable to replicate in vertebrates and their cells, offer new opportunities for vaccine development. These viruses are gaining increasing interest since they have the capability to suppress the replication of arboviruses when co-infected in insect cells. This indicates their potential use as biological control or as platforms for the development of vaccines and laboratory diagnostics [6,7].

Recent studies highlight the development of novel vaccines by creating viral chimeras using ISV’s. An ISV is used as the backbone and reading frames of primarily structural proteins of the related vertebrate pathogen are inserted in frame, which creates a virus unable to replicate in the vertebrate host but displaying the most important proteins for cellular uptake in humans. This review aims to present an update about the potential vaccines for arboviruses focusing on this new technology using ISVs.

## 2. Overview of Health Public Problems Caused by Arboviruses

Arboviruses have caused epidemics and outbreaks in many countries, leading to the illness and death of thousands of people. Some arboviruses are also endemic in certain areas. The arboviruses ZIKV, DENV, CHIKV, Yellow Fever virus (YFV), WNV and Japanese encephalitis virus (JEV) are most notable in terms of global importance [4]. In general, most infections of these viruses cause no or mild illness composed of general symptoms like fever, headache, and malaise [5].

DENV (*Flaviviridae*, *Flavivirus*), with four serotypes, DENV-1, DENV-2, DENV-3 and DENV-4, is an arboviral pathogen that causes the most cases in humans annually. Approximately 390 million cases of Dengue occur per year with 96 million presenting as mild to severe and fatal disease. Furthermore, studies indicate that 3.9 billion people are at risk of getting Dengue in at least 129 countries [8,9,10]. The disease is worldwide, affecting areas in the Americas, Africa, the Eastern Mediterranean, South-East Asia, Europe and the Western Pacific [10].

Before 2007, ZIKV (*Flaviviridae*, *Flavivirus*) had caused just sporadic cases of human disease mainly in Africa and Asia. Conversely, in 2007, ZIKV was responsible for an outbreak on the Island of Yap (Federated States of Micronesia, Oceania). After this, the virus spread to French Polynesia in 2013 and other countries and territories in the Pacific, causing outbreaks of bigger impact. Two years later, an epidemic of ZIKV in Brazil, initially associated with rash, was then associated with Guillain–Barré syndrome and microcephaly in newborns. Approximately 3500 cases of congenital abnormalities have been attributed to ZIKV in at least 86 countries and territories in the Americas, Africa, Asia, Oceania and more recently in Europe [11,12,13]. Many more cases of abnormal child development are likely still unattributed to maternal infection.

Although first reported in 1952 on Tanzania (East Africa), since 2004, CHIKV (*Togaviridae*, *Alphavirus*) has spread to at least 60 countries and territories on all different continents, causing human disease characterized mainly by fever and joint pain. The resultant chronic arthritis is frequently debilitating and may last months or years. Most patients recover well, however, the symptoms can become chronic or the disease can lead to death [14,15]. More recent outbreaks can be explosive, an example highlighted in Chad (north-central Africa) where 27,540 cases were report to the World Health Organization from July to September in 2020.

West Nile virus, a member of the family *Flaviviridae*, genus *Flavivirus*, is an arbovirus that infects birds, mosquitoes, human, horses and other mammals. Approximately 80% of infected people do not present with any clinical manifestations, however, the virus has caused outbreaks of neurological disease, including fatal outcomes in humans, in different areas in Americas, Europe, Africa, Middle East and Asia. Animals such as horses and birds can develop severe disease and subsequent death. When this virus encroaches, explosive outbreaks can occur; the North American outbreak caused over 30,000 human and 25,000 equine cases in its first five years of activity. There is no specific treatment and vaccines against WNV available for humans. Vaccines are only available for horses [16].

Though the above-mentioned viruses are the most known arboviruses, many other viruses belonging to this group are circulating and causing disease such as YFV, JEV, Rift Valley fever virus (RVFV), and Venezuelan equine encephalitis virus (VEEV). Certain arboviruses, such as the Bluetongue virus (BTV), can cause severe disease in ruminants, leading to major veterinary economic impact and disruption of local and global food supplies. In addition, there are other human pathogenic arboviruses considered at risk of emergence or reemergence. Two examples are the Mayaro virus (MAYV) and Oropouche virus (OROV), which circulate in some areas, such as South America, causing isolated outbreaks, but may emerge in new areas [4]. The possibility of emergence of arboviruses is driven by human travel, global commerce, urbanization, deforestation, global warming favoring vectors, and construction of man-made water bodies such as lakes of hydroelectric power plants that create mosquito habitat [4,5].

## 3. Currently Available Arbovirus Vaccines

Although arboviruses are causing diverse epidemics around the world, there are still few vaccines licensed at this time. According to the official list of licensed vaccines of US Food and Drug Administration (FDA) and WHO, there are vaccines available against DENV, YFV, WNV (horses), tick-borne encephalitis virus (TBEV) and JEV. There are also vaccines in development for CHIKV, ZIKV, and WNV (human) [17,18,19].

The yellow fever (YF) vaccine was the first vaccine developed for an arbovirus, more than 80 years ago. This vaccine has been key in the control of YF-induced hemorrhagic disease. However, irrespective of vaccine use, YF still emerges and reemerges cyclically in many areas. When this occurs in areas without immunization programs, deaths in nonhuman primates (NHP) and humans [17] are common.

The vaccine for DENV was recently licensed (Dengvaxia, CYD-TDV, Sanofi Pasteur) for protection against all four serotypes, however efficacy between serotypes is highly variable: DENV-1 (61.2%), DENV-2 (3.5%), DENV-3 (81.9%) and DENV-4 (90%). This vaccine was recommended for individuals from 9–45 years of age living in endemic areas with previous laboratory-confirmed infection. In 2017 Sanofi Pasteur indicated that their vaccine should be only used in persons with prior exposure, since immunologically-naive recipients were more at risk for hemorrhagic disease if infected after vaccination. Because of the restrictions and low efficacy for DENV-2, use has been complicated by the recommendation that serological screening should be performed to confirm previous exposure before administration of the vaccine [17,20,21].

Since the Zika pandemic in 2015, a race to develop a safe and effective vaccine against ZIKV was initiated and several candidate vaccines, including inactivated vaccines, whole-virus vaccines, live virus vaccines, subunit vaccines, and messenger RNA (mRNA), DNA, protein- and vector-based preparations, are in a variety of stages of testing including preclinical, and stage 1 and 2 clinical trials [22].

There are also promising vaccine candidates against CHIKV in clinical phases of development using different strategies such as virus-like particle (VLP) vaccines, live attenuated CHIKV, measles vector platforms, inactivated vaccines, subunit vaccines, chimeric vaccines, and nucleic acids [23,24,25,26].

## 4. Interaction between Insect-Specific Viruses and Arboviruses

The currently known ISVs are diverse and belong to many virus families including *Flaviviridae*, *Togaviridae*, *Rhabdoviridae*, *Reoviridae*, *Mesoniviridae*, *Peribunyaviridae*, *Phenuiviridae*, *Tymoviridae*, *Birnaviridae*, *Nodaviridae*, *Parvoviridae*, *Iridoviridae*, *Permutotetraviridae*, *Iflaviridae*, *Orthomyxoviridae*, *Totiviridae* and the proposed taxon *Negevirus* [5]. Phylogenetically, some ISVs, mainly those belonging to the viral families *Flaviviridae* and *Togaviridae*, are closely related to pathogenic arboviruses. Meanwhile, other ISVs are more related to plant viruses such as those included in the family *Tymoviridae* and the taxon *Negevirus* [6,7]. ISVs belonging to the viral families *Flaviviridae* and *Togaviridae* are more commonly detected, characterized and studied, with the need to study the ISVs from other viral families including, their transmission mechanisms, evolutive process and maintenance in nature.

Studies have shown that certain ISVs modulate arbovirus replication and the vector competence of mosquitoes. For example, the insect-specific flavivirus Nhumirim (NHUV) (*Flaviviridae*, *Flavivirus*) suppressed the replication of the flaviviruses WNV, Saint Louis encephalitis virus (SLEV), and Japanese encephalitis virus (JEV) in C6/36 cells up to a million-fold and 10,000-fold reduction in titers [27]; furthermore, the prior inoculation of *C. quinquefasciatus* mosquitoes by NHUV, significantly decreased the competence of those mosquitoes of transmitting WNV [28].

Another example is the insect-specific alphavirus Eilat (EILV) (*Togaviridae*, *Alphavirus*), which demonstrated suppression of the replication of arboviruses of the same genus, including Sindbis virus (SINV), VEEV, Eastern equine encephalitis virus (EEEV), Western equine encephalitis virus (WEEV) and CHIKV, reducing titers by 10–10,000-fold. In vivo experiments showed that *A. aegypti* mosquitoes previously infected with EILV presented delayed dissemination of CHIKV [29]. Another study showed that the EILV resulted in both homologous and heterologous interference with other alphaviruses [29].

## 5. Perspectives for the Prevention of Disease Caused by Arboviruses

New strategies for the prevention of disease caused by arboviruses have been developed, including the possible use of mosquito saliva proteins (MSP) as a universal vaccine against arboviruses. Insect saliva facilitates the transmission of pathogens, like viruses, making the host infection process easier and, thus, it is theorized that anti-MSP vaccines may also block virus transmission [30]. The explanation is that the mosquito bites promote inflammatory responses at the inoculation site, recruiting defense cells such as neutrophils, retaining the virus at the injection site resulting in macrophage recruitment, and infection, allowing virus replication. This inflammatory response, promoted by the mosquito bite, is one of the determining factors for the clinical severity of disease [31]. Based on this, mosquito saliva vaccines may modify the host response to the insect saliva, avoiding spread of the infection and effectively blocking the transmission [30].

Another potential strategy for the development of vaccines against arboviruses is the use of ISVs as platforms. This is a promising avenue for vaccine development due to the growing number of ISVs recently identified and the advancements in genome sequencing of the mosquito microbiome. Recent studies demonstrate the possible application of ISVs as biological control armaments, since these viruses suppress the replication of certain arboviruses. Furthermore, a novel application for ISV is their use in laboratory diagnostics, since chimeras of the ISV EILV and alphaviruses (CHIKV, VEEV, EEEV) present highly specific antigens for ELISA. Within the laboratory setting, work with ISVs have other advantages as safety, lower costs, efficiency, and the development of high titers in cell culture in short times leading to less need for concentration, purification or inactivate preparations [32,33].

### 5.1. ISVs as Platforms for Vaccine against Arboviruses

The ISVs naturally infect mosquitoes and phlebotomine sandflies, replicating in their cells, in vivo and *in vitro*, but are unable to replicate in vertebrates and their cells [6,7]. Based on the similarity of some ISVs with arboviruses, vaccine candidates using these viruses were proposed (Table 1). This strategy offers increased safety due to the inability to complete replication in vertebrate cells compared, for example, with attenuated vaccines where the risk of disease exists from partial inactivation, mainly in immunocompromised individuals. Also, when compared with inactivated vaccines, these constructs do not need chemical or physical inactivation, which often decreases or even abrogates immunogenicity and antigenicity of the wild-type virus [34]. The ISV-based vaccines are developed using recombinant technology to generate chimeras (Figure 1).

A vaccine against CHIKV (*Togaviridae*, *Alphavirus*) was developed using the insect-specific alphavirus Eilat (EILV) (*Togaviridae*, *Alphavirus*) as the platform [34]. A chimeric virus was designed with EILV cDNA clone and the CHIKV structural proteins E1, E2 and C. The chimeric virus demonstrated successful replication, similar to CHIKV, in their early stages (attachment and entry to viral RNA delivery) in vertebrate cells, but were defective in productive replication. The chimeric vaccine was inoculated in laboratory animals (rodents) and the genome of neither virus was detected, pointing to the absence of complete replication [34].

This EILV/CHIKV chimeric vaccine demonstrated fast, robust and durable immunogenicity. The study data showed that a single dose of this vaccine was protective in two different mouse models and an NHP model generating fast (within 4 days) and long-lasting (more than 290 days) neutralizing antibodies [34]. Complete protection was demonstrated in the mouse experiments and fever and viremia was prevented in the NHP model. [34]. Comparatively, when the EILV/CHIKV chimera was inactivated and compared with the live chimera, immunity was less effective; nonetheless, the inactivated virus was immunogenic and decreased the mortality of the animals, though not able to eliminate disease manifestations in certain mouse models.

A recent study investigated the immunogenicity of the EILV/CHIKV chimeric vaccine in mice. The results demonstrated that a single dose of the chimera promoted high levels of CHIKV-specific IgM and IgG antibodies, memory B cell and CD8^+^ T cell responses. The EILV/CHIKV vaccine induced antiviral cytokines and antigen presenting cells (APCs) in vivo, but did not induce APCs alone in vitro. In summary, the EILV/CHIKV vaccine promoted strong innate and adaptive immunity in mice [35].

The EILV platform was also used to create Eastern equine encephalitis virus (EEEV) and VEEV chimeras. Animals were vaccinated with a single dose of monovalent or multivalent EILV/EEEV and EILV/VEEV live chimeras. Neutralizing antibodies were generated and mortality was prevented in mice when challenged by the wild-type viruses 70 days later. In mice inoculated with a single dose of monovalent vaccine, 50% of the animals were protected by 6 days post vaccination [36].

Pathogens of the genus *Flavivirus*, DENV, YFV, ZIKV, WNV and JEV were combined with recently discovered ISV called Binjari virus (BINJV) as the viral backbone. The BINJV structural protein genes (prME) were exchanged by those genes of each of the aforementioned arboviruses using a modified circular polymerase extension reaction (CPER) methodology and were confirmed to be structurally and immunologically similar to the parental arboviruses. These constructs replicated well in mosquito cells, but as expected were replication-defective in vertebrate cells. [37].

This new vaccine technology is an interesting pipeline for rapid response to emergent pathogenic flaviviruses, since the whole process of production was fast, lasting two to three weeks [37].

The chimeric vaccine against ZIKV, named BINJV/ZIKV-prME, was inoculated into IFNAR -/- mice with a single dose of 2 or 20 µg. A significant immune response was observed and characterized by the production of ZIKV-specific IgG1 and IgG2c, total IgG and neutralizing antibodies. Higher responses were observed using the 20 ug vaccine, but both dosages decreased viremia to below detection levels after challenge with ZIKV [37].

The chimeric antigens (BINJV/WNVKUN-prME, BINJV/DENV-prME, BINJV/ZIKV-prME, BINJV/YFV-prME and BINJV/JEV-prME) were also used to develop diagnostic immunoassays such as ELISA and MIA (multiplex microsphere immunoassay) [37].

The use of an experimental BINJV/WNVKUN-prME vaccine (against the Kunjin strain of WNV) for animals living outdoors in farms under minimal biocontainment conditions was approved by the Australian authorities of regulation aiming to avoid WNV-KUN infections [37].

#### 5.1.1. Advantages and Disadvantages of ISV-Based Vaccines

As previously described, ISV-based vaccines present advantages such as safety due to the absence of replication of the chimeric virus in vertebrates. This is as opposed to attenuated vaccines based solely on wild-type viruses, which may revert to virulence. Production of these vaccines can be performed at a lower biosafety level, making the process cheaper and safer. Issues of incomplete inactivation during production is not an issue since there is no inactivation. Finally, conformationally dependent epitopes similar to wild-type are conserved.

Despite all those advantages, there are safety concerns regarding the use of ISV-based vaccines in humans. Production of these vaccines is likely best accomplished in C6/36 and C7/10 (*A. albopictus* larvae cells) cells, since the chimeras grow to high titers in insect cells. However, hypersensitivity may be an issue for antigens produced in mosquito cells. In the 1980s, interest in the use of C6/36 cells grew because of the high titer of DENV in these cells, which was attractive for DENV vaccine production. Because there were concerns about the safety of mosquito larvae products in human applications, safety studies using a C6/36 sham vaccine were performed in twelve human volunteers. While none of those volunteers had reactions to intramuscular injections, three developed immediate reactions after intradermal injection. The remaining nine other participants then received another subcutaneous injection and five out of nine developed delayed-type hypersensitivity (DTH) reactions (12 h after) at the site of the injection [38].

Recently published work demonstrated no safety concerns in mice inoculated with the EILV/CHIKV chimeric vaccine prepared in C7/10 cells. In this study, the chimeric virus was purified using a two-step methodology, first by affinity chromatography (Cellufine^®^ Sulfate, JNC Co, Tokyo Japan) followed by ultracentrifugation in a discontinuous sucrose gradient. The purified antigen did not induce any more responses than minor injection site inflammation in mice [35].

#### 5.1.2. Perspectives for the Use of ISV-Based Vaccines

The vaccines against arboviruses such as ZIKV, WNV, CHIKV, EEEV and VEEV, produced using ISVs and recombinant technology have shown satisfactory responses with a single dose; also, they offer more safety due to the ISVs’ inability to replicate in vertebrate cells. Given the results of the study conducted by Adam and colleagues [35] showing that reactivity can be minimized by enhanced purification, this reinforces the possible future use of ISV-based vaccines in humans. More studies are needed to robustly characterize the immune response of humans to ISV-based vaccines. Additionally, it is extremely important to perform clinical trials in humans to fully investigate immediate DTH reactions and the efficacy of these vaccines to protect against arbovirus infections.

Even though several ISVs have been isolated in the last years, more exploration is needed for their potential uses, especially as vaccine platforms. Additionally, despite the many advantages of ISV-based vaccines, mainly the inability of these viruses to replicate in vertebrates, very few studies have been published about the development of ISV-based vaccines and at this moment, there is only one already licensed and in use to protect animals against WNV infection.

Although several novel vaccine candidates such as plant-based vaccines proved difficult to develop and only generated limited interest in public participation in clinical trials, the current acceptance by humans for novel platforms during the coronavirus pandemic may well lead to industry investment. Presented with the excellent data now being generated in the recent literature, there is more opportunity to bring much needed safe and immunogenic vaccines to market to protect against arboviruses.

#### 5.1.3. Strategies of Research

Here we present an overview of the use of ISV’s as potential vaccine candidates. The search strategy for this narrative review is indicated in Table 2.

## 6. Conclusions

Arboviruses cause serious problems for public health systems around the world, causing disease and death of humans and animals. At this moment, there is no specific treatment, and there are few licensed vaccines to fight these viruses. In fact, some of the available vaccines present restrictions, making it ever more important to develop new vaccines with a broader spectrum of protection.

ISVs are promising tools for the biological control of arboviruses, vector control, vaccines and diagnostic platform development. In view of the significant number of cases of disease, and the possibility of emergence of these viruses in new areas, the development of new and improved vaccines against arboviruses is crucial to protect the global population of humans and animals against illness and death caused by these viruses. ISVs offer an important opportunity for vaccine development and placement of currently developed ISV chimeras into further clinical development. Further studies in this area using other ISVs and arboviruses are also encouraged.

## Figures and Tables

**Figure 1 vaccines-09-00263-f001:**
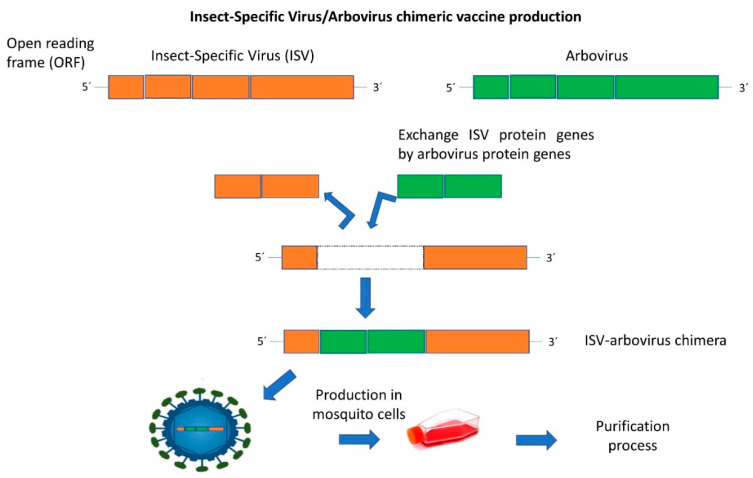
Schematic description of the development and production of insect-specific virus-based chimeric vaccines against arboviruses.

**Table 1 vaccines-09-00263-t001:** Insect-specific virus-based vaccines developed against arboviruses.

Arboviruses	ISV	Strategy	Titers In Vitro	Animal Challenge	Reference (Year)
Chikungunya (CHIKV)	Eilat virus (EILV)	Development of a chimeric virus with EILV cDNA clone and the CHIKV structural proteins: E1, E2 and C	10^10^ PFU/mL	C57BL/6 mouse, A129 IFNα/βR−/− mice, nonhuman primates (NHPs) (cynomolgus macaques)	2017
Eastern equine encephalitis virus (EEEV), Venezuelan equine encephalitis virus (VEEV)	Eilat virus (EILV)	Development of a chimeric virus with EILV cDNA clone and the EEEV and VEEV structural proteins: E1, E2 and C	10^6^ to 10^8^ PFU/mL	CD-1 mice	2018
Dengue (DENV), Yellow Fever (YFV), Zika (ZIKV), West Nile (WNV), Japanese Encephalitis (JEV)	Binjari virus (BINJV)	Development of a chimeric virus using BINJV as backbone for structural protein genes (prME) of DENV, YFV, ZIKV, WNV and JEV	Up to 10^9.5^ cell culture infectious dose/ml	Murine IFNAR−/− mouse models	2019

**Table 2 vaccines-09-00263-t002:** Summary of the search strategy, sources employed, timeframe and citation number.

Search Strategy	(Insect Specific Viruses Vaccine) and (Insect Specific Viruses, Vaccines, Arboviruses)	(Insect Specific Viruses, Vaccines) and (Insect Specific Viruses), Vaccines)	“Insect-Specific Viruses” “Vaccine” Not Review	“Insect-Specific Viruses” “Vaccine” “Arbovirus” Not Review
Source	Pubmed.gov	Pubmed.gov	Google Scholar	Google Scholar
Time Frame	1971–2020	1971–2020	1970–2021	1970–2021
Citation Number	64	824	207	154

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
