# Peer review of "Perspectives on New Vaccines against Arboviruses Using Insect-Specific Viruses as Platforms"

_vaccines, 2021, doi:10.3390/vaccines9030263_

Round 1

Reviewer 1 Report

The authors here review the public health importance of arboviruses, current vaccines and other preventative efforts, interaction with ISVs, and then discuss ISV platforms for arbovirus vaccines. While that discussion is brief, there are not many current examples to explore so this discussion is sufficient. ISVs present a very promising No major revisions are requested, though minor editing is needed and some additional points of discussion are warranted.

While the authors describe the need for and promise of ISVs along with current examples, the possible challenges of this work should also be addressed. Namely, there has been some small concern that vaccines produced on mosquito cells lines may induce a hypersensitivity response (Human hypersensitivity to a sham vaccine prepared from mosquito-cell culture fluids, Scott RMN et al, 1984, Journal of Allergy and Clinical Immunology). A recent study (Optimized production and immunogenicity of an insect virus based chikungunya virus candidiate vaccine in cell culture and animal models, Adam A et al, 2021, Emerging Microbes and Infections) has helped to ease these concerns and also gives greater detail on the immune response to the Eilat-Chikv vaccine.

Author Response

Thank you! We appreciate your suggestions. We reviewed the English. We included the required information on the manuscript (highlighted in yellow), specifically in the topics:

5.1. ISVs as platforms for vaccine against arboviruses;

5.1.1. Advantages and Disadvantages of ISV-based vaccines;

5.1.2. Perspectives for the use the ISV-based vaccines.

Reviewer 2 Report

This manuscript describes the possibility of using insect-specific viruses (ISVs) as platforms for vaccine production. While this platform has not been fully explored in humans, this review may provide some insights to justify the use of ISVs for vaccine development. The review, however, was too general and did not succinctly communicate the advantages and disadvantages of exploring ISVs for vaccine development. I would thus recommend the authors to include topic sections on the advantages and disadvantages of using ISVs, as well as strategies and perspectives of how we can potentially exploit ISVs for vaccine development. In addition, it will be extremely useful to include at least one table and figure to summarize the topic review.

Author Response

Thank you! We appreciate your suggestions. We reviewed the English. We included the required information on the manuscript (highlighted in yellow), specifically in the topics:

5.1. ISVs as platforms for vaccine against arboviruses;

5.1.1. Advantages and Disadvantages of ISV-based vaccines;

5.1.2. Perspectives for the use the ISV-based vaccines.

We also included tables and one figure as you recommended.

Reviewer 3 Report

Estimated Authors,

Estimated Editors,

I've read with interst the present paper, a narrative review on new vaccines against arboviruses build-up through platforms based on Insect-specific viruses. 

Authors have performed a comprehensive review of available evidence, that was reported in a relatively simple and straightforward way.

In my opinion, the overall quality of this paper is compatible with the requirements of Vaccines. I've only a minor recommendation, i.e. include two summary table:

1) Table 1, including a summary of your research strategy, the sources you employed, and the timeframe of your inquiry;

2) Table 2, summarizing the data on the ISV or ISV-based platform that may be employed for new human vaccines, with corresponding characteristics 

Author Response

Thank you very much! We appreciate your comments and suggestions. We reviewed the English. We included the required information on the manuscript (highlighted in yellow), specifically in the topic:

5.1. ISVs as platforms for vaccine against arboviruses;

We included one table (Table 1) summarizing the data on the ISV or ISV-based platform that may be employed for new human vaccines, with corresponding characteristics, as you recommended. We think that this will enrich our paper. Regarding to the other table you suggested, our review is not a systematic review, but it is a literature/general review. Even though, we included this table who we named Table 2 at the topic 5.1.3. Strategies of research.

Round 2

Reviewer 2 Report

The authors have fully addressed my questions